# Screening microbially produced Δ⁹-tetrahydrocannabinol using a yeast biosensor workflow

William M. Shaw[1,2,3,4], Yunfeng Zhang[5], Xinyu Lu[3,4], Ahmad S. Khalil [1,2,6], Graham Ladds [7], Xiaozhou Luo[5] & Tom Ellis [3,4] ✉

Microbial production of cannabinoids promises to provide a consistent, cheaper, and more sustainable supply of these important therapeutic molecules. However, scaling production to compete with traditional plant-based sources is challenging. Our ability to make strain variants greatly exceeds our capacity to screen and identify high producers, creating a bottleneck in metabolic engineering efforts. Here, we present a yeast-based biosensor for detecting microbially produced Δ⁹-tetrahydrocannabinol (THC) to increase throughput and lower the cost of screening. We port five human cannabinoid G protein-coupled receptors (GPCRs) into yeast, showing the cannabinoid type 2 receptor, CB2R, can couple to the yeast pheromone response pathway and report on the concentration of a variety of cannabinoids over a wide dynamic and operational range. We demonstrate that our cannabinoid biosensor can detect THC from microbial cell culture and use this as a tool for measuring relative production yields from a library of Δ⁹-tetrahydrocannabinol acid synthase (THCAS) mutants.

Over recent years, our perception of cannabinoids has shifted from an illicit drug to a promising new prescription medicine for conditions such as epilepsy and treating the symptoms of multiple sclerosis and chronic pain[1,2]. Although a great deal of research remains to uncover the full therapeutic benefits of this class of natural products, there is considerable promise for treating numerous other conditions[3], with hundreds of clinical trials currently underway. A growing excitement combined with relaxed regulation has led to an explosion in the number of companies dedicated to the production of cannabinoids[4].

However, botanical extract from cannabis remains the principal source of these compounds[5]. Δ⁹-Tetrahydrocannabinol (THC) and cannabidiol (CBD), the two most widely studied and clinically approved cannabinoids, are normally only found in the small outgrowths in the flowers of female plants, known as trichomes, meaning most of the plant is wasted biomass[5–7]. Additionally, purification can be very expensive and often leads to a complex mixture of cannabinoids. It is time consuming and expensive to obtain plants with the desired composition using traditional breeding, with this issue being further complicated by rarer cannabinoids which may be present in trace amounts[5,8,9].

To address this issue, researchers are turning to bioengineering by genetically altering cannabis to make cannabinoids in other areas of the plant and/or manipulating metabolic flux, thereby increasing overall yields of the desired compound[6,10,11]. Alternatively, synthetic biologists have begun engineering microbes, such as baker's yeast and *Escherichia coli*, by introducing entire biosynthetic pathways to produce cannabinoids from cheap feedstocks[6,9]. Biosynthesis has also been demonstrated in cell-free systems, achieving yields up to 0.5 g/L[8].

[1]Biological Design Center, Boston University, Boston, MA 02215, USA. [2]Department of Biomedical Engineering, Boston University, Boston, MA 02215, USA. [3]Department of Bioengineering, Imperial College London, London SW7 2AZ, UK. [4]Imperial College Centre for Synthetic Biology, Imperial College London, London SW7 2AZ, UK. [5]Center for Synthetic Biochemistry, Shenzhen Institute of Synthetic Biology, Shenzhen Institute of Advanced Technology, Chinese Academy of Sciences, Shenzhen 518055, China. [6]Wyss Institute for Biologically Inspired Engineering, Harvard University, Boston, MA 02115, USA. [7]Department of Pharmacology, University of Cambridge, Cambridge CB2 1PD, UK. ✉e-mail: t.ellis@imperial.ac.uk

Microbial and cell-free production has the added benefit of producing purer extracts of the target molecule by introducing specific combinations of enzymes and precursor metabolites[5]. This is a particularly advantageous approach for obtaining rare cannabinoids and analogues not seen in nature, which may have unknown useful properties[8,9].

While microbial and cell-free production are enticing, improving yields and scaling production to compete with traditional plant sources is challenging and costly[6,12]. Recent reduction in the cost of DNA synthesis and improved genome engineering tools means our ability to build strains now vastly outweighs our capacity to test for high producers, creating a bottleneck in the metabolic engineering development cycle[13,14]. Quantification methods, such as liquid chromatography-mass spectrometry (LC-MS), lack the required throughput and are prohibitively expensive for smaller research groups and companies, both in terms of the equipment and expertise required[13,14].

Biosensors have emerged as a cheap and powerful tool for the detection and quantification of various metabolites[15]. Linking a responsive element, such as a cell-surface receptor, to the expression of a reporter gene, such as green fluorescent protein (GFP) or a selection marker, allows the indirect measurement of a metabolite using a medium-throughput plate reader or growth-based assay.

Although cell-surface receptors are limited to sensing molecules outside of the cell (precluding the linkage of production with biosensing and access to high-throughput selection), decoupling production from biosensing has benefits. Firstly, the preparation of microbial samples can ensure metabolite concentrations are always within the linear range of the biosensor, allowing the accurate reporting of titres as the producer strain improves. Sample preparation also provides an opportunity to perform additional conversion steps towards the final product in vitro, if required. Secondly, random, genome-wide mutagenesis strategies can be applied to the producer strain without affecting biosensor function or creating cheater cells. Finally, extracellular production can be separated from intracellular production if the secreted yield of a metabolite is of interest.

To facilitate the creation of biosensors able to detect a wide range of biological inputs, our group and others have built platforms for porting mammalian G protein-coupled receptors (GPCRs) into the widely used yeast, *Saccharomyces cerevisiae* (*S. cerevisiae*), and coupling activation to a measurable output[16–18]. GPCRs can detect a broad range of ligands and stimuli, and yeast biosensors have now been developed for the detection of microbially produced serotonin, melatonin, and fatty acids, for the presence of fungal pathogens, and recently were used as an engineered probiotic for the detection and treatment of inflammatory bowel disease in mice[16,18–21].

Here, we present a GPCR-based biosensor that is sensitive to a range of cannabinoids by expressing the human cannabinoid type 2 receptor, CB2R, in yeast. Using this biosensor yeast, we demonstrate the detection of microbially-produced $\Delta^9$-tetrahydrocannabinol (THC) and use the living sensor as a tool for measuring relative production yields from a library of $\Delta^9$-tetrahydrocannabinolic acid synthase (THCAS) mutants.

## Results

### Developing a yeast GPCR-based cannabinoid biosensor

To create a biosensor for cannabinoids, we built on our previously reported platform for developing highly tuned GPCR-based biosensors in yeast[16]. In brief, this platform consists of an extensively modified strain of *S. cerevisiae* with a minimised pheromone response pathway (yWS677). 15 genes were deleted to increase signalling, prevent unwanted cell cycle arrest and mating gene expression, and provide a null background for GPCR expression. This strain is complemented by a modular genetic toolkit for expressing heterologous GPCRs, a library of Gα proteins for receptor-pathway coupling, and a synthetic transcription factor to redirect the pathway response to a

synthetic promoter. Placing GFP downstream of the synthetic promoter, for example, provides a measure of GPCR activation using cell fluorescence. This can be used to report on external concentrations of the GPCR-specific ligand(s) for biosensing purposes.

We first selected a panel of GPCRs from the human genome that are known to have sensitivity to cannabinoids. The first two obvious choices were the cannabinoid type 1 and 2 receptors (CB1R and CB2R) that function in the endocannabinoid system (ECS), regulating a diverse range of physiological functions, such as appetite, pain sensation, and inflammation[22,23]. These two receptors are also modulated by cannabinoids, such as THC and CBD, with the CB1 receptor being responsible for the psychoactive effects in the central nervous system and the CB2 receptor having less well understood consequences on the immune system, such as anti-inflammatory effects[22,23]. Additionally, the binding of cannabinoids has been identified in several other GPCRs, including GPR18, GPR55, and GPR119, which may also play a role in the ECS[24,25]. Previous reports have shown all of these receptors should express and functionally couple in yeast[17,26–28].

We codon optimised the open reading frames (ORFs) of the five human GPCRs for expression in yeast and cloned them into a vector under the control of the strong constitutive *CCW12* promoter using the Yeast MoClo Toolkit[29]. We then introduced the receptors into the yWS677 GPCR chassis strain alongside a library of 12 Gα proteins, consisting of the wildtype yeast Gα (Gpa1), 10 yeast-mammalian chimeric Gα's, where the last 5 amino acids of Gpa1 are substituted for the mammalian equivalents, and a truncated Gpa1 protein control (tGpa1), where the last 5 amino acids of Gpa1 have been deleted to prevent receptor-G protein signalling[16] (Fig. 1a). In all conditions the output of the pathway is mediated by the synthetic LexA-PRD transcription factor driving expression of GFP from the LexA(6x)-*pLEU2m* promoter, which we previously demonstrated to produce a high fold-change in reporter output[16].

Next, we tested all 60 strains in the presence and absence of 10 μM of their endogenous agonists (CB1R and CB2R, 2-arachidonoylglycerol (2-AG)[22]; GPR18, *N*-arachidonylglycine (NaGly)[30]; GPR55, lysophosphatidylinositol (LPI)[31]; GPR119, oleoylethanolamine (OEA)[28], Fig. 1b), and measured the response in a plate reader (Supplementary Fig. 1). We then calculated the fold-change in GFP expression, subtracting the change seen in the control tGpa1 strains to discount any receptor-independent effects on reporter output caused by the ligand (Fig. 1c). Contrary to previous reports[26–28], only the CB2 receptor showed reporter activity with the addition of ligand, where we saw a significant fold-change across most Gα variants. To bolster heterologous GPCR activity in yeast, changes to the underlying yeast strain or conditions, such as pH and growth temperature, have been shown to improve signalling[21]. However, as CB2R demonstrated a high fold change in standard conditions, we focused on this receptor for biosensing.

We re-characterised the CB2R/Gα strains more thoroughly using flow cytometry, also including the endocannabinoid anandamide (AEA), for higher resolution of the pathway response and to confirm receptor activation using an additional endogenous ligand[32] (Fig. 1d). This identified the wildtype yeast Gpa1 as the best performing Gα, demonstrating up to 60-fold change in GFP expression in the presence of 10 μM 2-AG (Fig. 1e). Moving forward with the CB2R/Gpa1 biosensor strain (CB2 biosensor, yWS2345), we characterised the dose-response with 2-AG and AEA over a range of concentrations. AEA presented a more typical sigmoidal dose-response (Hill slope = 0.8), with a greater than 2-log concentration range between 10 % and 90 % of the response (operational range) and a 53-fold maximum change in GFP fluorescence (dynamic range) (Fig. 1f). 2-AG presented a more abnormal, non-sigmoidal dose-response, due to a receptor-independent effect from the ligand, discussed later (Supplementary Fig. 2). No change in GFP expression was seen in the CB2R/tGpa1 control strain (yWS2356) with either ligand, showing that reporter upregulation was CB2 receptor dependent. As the CB2 biosensor demonstrated low basal activity and

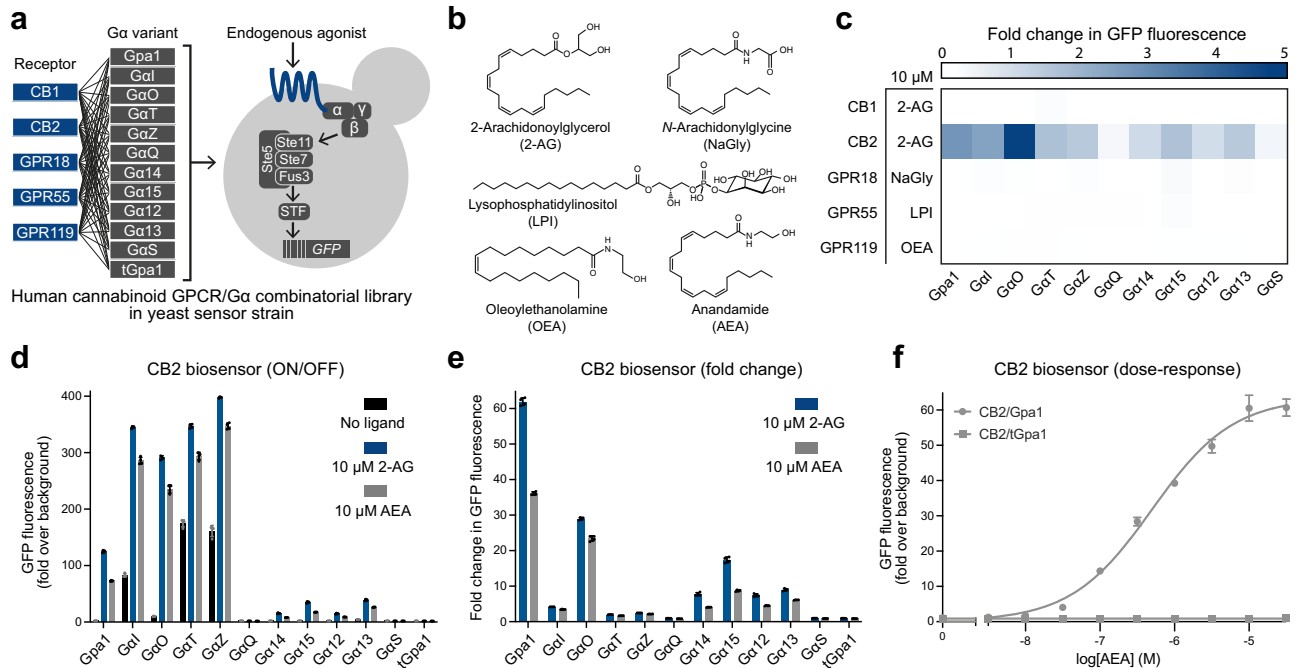

**Fig. 1 | Functional coupling of human cannabinoid-responsive G protein-coupled receptors in yeast. a** Combinatorial library of five human cannabinoid receptors in combination with 12 Gα protein variants introduced into the yeast GPCR chassis strain (yWS677) with a GFP pathway reporter output. **b** Chemical structure of 2-arachidonoylglycerol (2-AG), N-arachidonylglycine (NaGly), lysophosphatidylinositol (LPI), oleoylethanolamine (OEA), and anandamide (AEA). **c** Maximum fold change in GFP reporter output in the GPCR/Gα combinatorial strain library with the addition of 10 μM of 2-AG (CB1R and CB2R), NaGly (GPR18), LPI (GPR55), and OEA (GPR119). Experimental measurements are GFP levels per cell as determined by a plate reader and shown as the mean from four biological replicates. Fold change in GPCR/tGpa1 was subtracted from all data to discount for receptor-independent ligand effects. **d** CB2R/Gα library response to 10 μM 2-AG (blue), AEA (grey), and no ligand (black). **e** Maximum fold change in GFP expression of the CB2R/Gα library in response to 10 μM of 2-AG (blue) and AEA (grey). **f** AEA dose-response curves with the CB2R/Gpa1 biosensor and the CB2R/tGpa1 control. Experimental measurements are GFP levels per cell as determined by flow cytometry and shown as the mean ± SD from four biological replicates. Curves were fitted using GraphPad Prism variable slope (four parameter) nonlinear regression fit.

a wide dynamic and operational range, no further optimisations were made.

## Characterising the CB2 biosensor response to cannabinoids and their precursors

Next, we set out to explore the dose-response profiles of cannabinoids and their precursors on the CB2 biosensor in the context of yeast cannabinoid production, as reported in Luo et al.[9] (Fig. 2). In this pathway, the immediate precursors, olivetolic acid (OA) and geranyl pyrophosphate (GPP), are combined to create the cannabinoid gateway precursor, cannabigerolic acid (CBGA). CBGA can then be diversified by various synthases, to create cannabinoids commonly found in cannabis, including cannabidiolic acid (CBDA), Δ⁹-tetrahydrocannabinolic acid (THCA), and cannabichromenic acid (CBCA). The non-enzymatic decarboxylation of these molecules by heat then produces the more familiar cannabinoid forms used for their therapeutic effects, such as cannabigerol (CBG), cannabidiol (CBD), Δ⁹-tetrahydrocannabinol (THC), and cannabichromene (CBC).

We first tested the CB2 biosensor response to OA and GPP, in the presence and absence of 1 μM AEA (approximately half-maximal effective concentration, $EC_{50}$), to determine antagonistic and agonistic properties, respectively (Supplementary Fig. 3a). No activation was seen in the absence of AEA, but a 31 % reduction and 29 % increase in respective GFP expression was seen in the presence of 1 μM AEA, with 10 μM OA and GPP provided (Fig. 2a). Similar changes were not seen in two controls; when using the MTNR1A receptor (which is highly specific for its cognate ligand, melatonin[16]), and when having constitutive GFP expression (Supplementary Fig. 4a). Thus, the OA and GPP effects on signalling are specifically due to action on the CB2 receptor.

We next performed a dose-response experiment with CBGA over a range of concentrations and revealed this molecule to be a potent

antagonist to the CB2 biosensor, as it completely inhibited the response to AEA when given at 10 μM (Fig. 2b). Similar antagonism was also seen with the other acid cannabinoids, CBDA, THCA, and CBCA, with varying half-maximal inhibitory ($IC_{50}$) values (Supplementary Fig. 3b). As before, no major changes were seen with the control strains, showing that these are receptor-dependent effects (Supplementary Fig. 4a).

Finally, we tested the dose-response of the CB2 biosensor with the therapeutically relevant decarboxylated cannabinoids, which revealed a diverse range of responses (Fig. 1c and Supplementary Fig. 3c). CBG was a weak agonist, CBD was a moderate antagonist, and THC and CBC were strong agonists (Fig. 2c). This matches well with the reported effects of THC and CBD in humans, where CBD is an antagonist of THC receptor activation, modulating its effects[33]. In all cases, except CBG, the response to the ligands was predominantly receptor dependent (Supplementary Fig. 4a). Although no changes to maximum growth rate were seen (Supplementary Fig. 4b), CBG resulted in a large reduction in receptor-independent GFP expression, highlighting the known antimicrobial effects of cannabinoids and their direct action on membranes due to lipophilic structures[34,35]. Care should be taken when sensing cannabinoids with the CB2 biosensor to untangle receptor-dependent and -independent responses.

Taken together, the CB2 biosensor was able to measure the cannabinoids, cannabigerol, cannabidiol, Δ⁹-tetrahydrocannabinol, and cannabichromene over wide concentration ranges, albeit with a broad diversity of responses. This natural receptor promiscuity and mixed ligand behaviour, while relevant in context of the human body, complicates the application of a CB2 biosensor for the measurement of cannabinoids from microbial cultures, as these will typically consist of a complex mixture of cannabinoids and their precursors which will obscure direct quantification of any individual cannabinoid species.

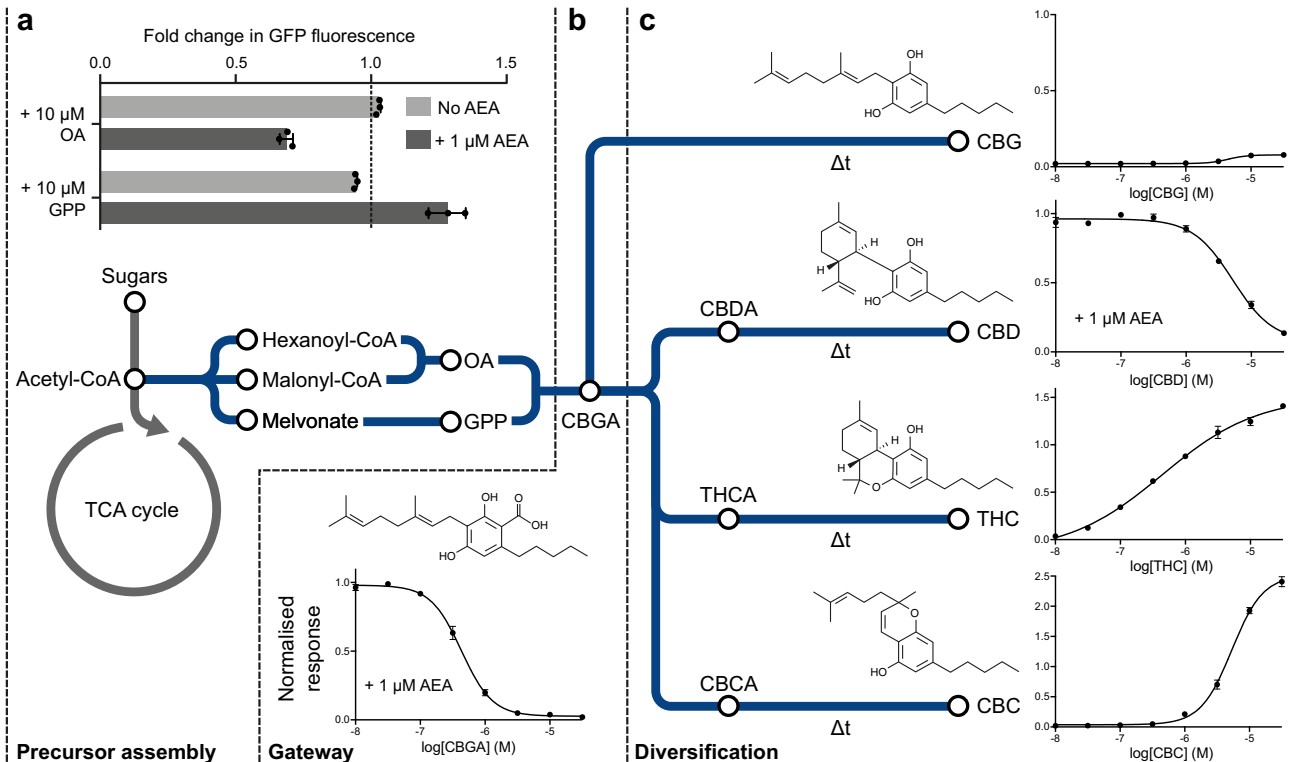

**Fig. 2 | Sensing cannabinoids and their precursors. a–c** Metabolic pathway for the complete biosynthesis of cannabinoids in yeast from sugar feedstocks, as demonstrated in Luo et al.[9]. **a** Fold change in CB2 biosensor expression after addition of 10 μM olivetolic acid (OA) and geranyl pyrophosphate (GPP) in the presence (dark grey) and absence (light grey) of 1 μM anandamide (AEA). **b** Cannabigerolic acid (CBGA) dose-response curve in the presence of 1 μM AEA with the CB2 biosensor. **c** Cannabigerol (CBG), cannabidiol (CBD), Δ9-tetrahydrocannabinol (THC), and

cannabichromene (CBC) dose-response curves with the CB2 biosensor. The CBD dose-response curve is in the presence of 1 μM AEA. Experimental measurements are GFP levels per cell as determined by flow cytometry and shown as the mean ± SD from three biological replicates. All data are normalised to the no ligand and 1 μM AEA CB2 biosensor response. Curves were fitted using GraphPad Prism variable slope (four parameter) nonlinear regression fit.

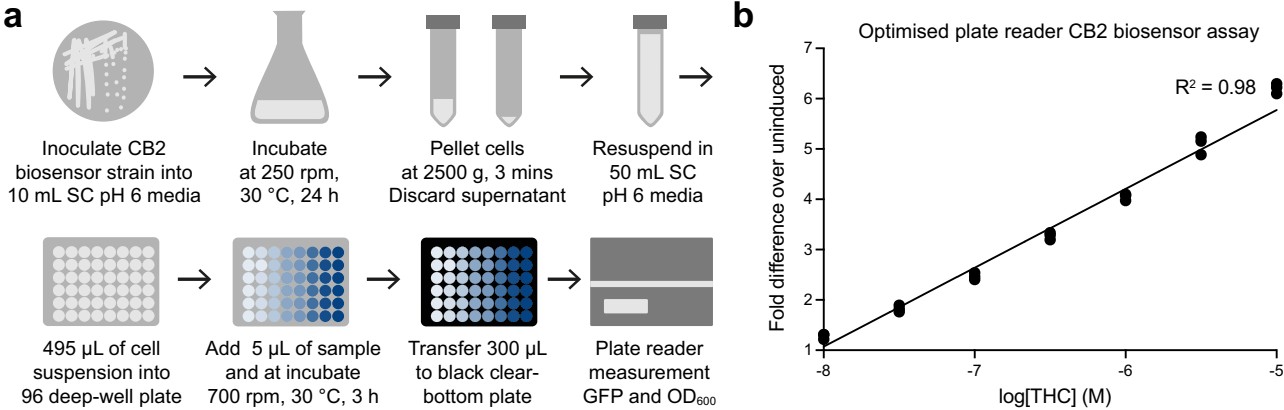

**Fig. 3 | Optimised CB2 biosensor plate reader assay. a** Optimised workflow for CB2 biosensor strain preparation, sample incubation, and plate reader measurement of 96 samples. Throughput can be linearly scaled with starting culture volume. **b** THC dose-response curve using the optimised CB2 biosensor plate reader assay over the linear range of GFP output (0.01–10 μM of THC). Data was

normalised to the uninduced CB2 biosensor response (1). Experimental measurements are GFP levels per cell as determined on a plate reader and shown as the individual values from three biological replicates. A straight-line curve was fitted using GraphPad Prism linear regression fit, $R^2 = 0.98$.

## An optimised plate reader assay for medium-throughput screening of THC

For screening the production of cannabinoids from microbial culture, the carboxylated forms that can be produced by enzymatic steps alone would not be practical, as the strong antagonistic response they produce could not be differentiated from the precursor, CBGA. Instead, the final decarboxylated cannabinoids could be used, as they all

present unique profiles that could be differentiated from CBG (which will be present from the conversion of unused CBGA). This would therefore require the decarboxylation of samples after extraction, adding one additional preparation step over LC-MS quantification. However, this step, which involves drying and heating samples[36], is scalable, low cost, and also represents the final commercial product, validating the ultimate amount of a cannabinoid that can achieved

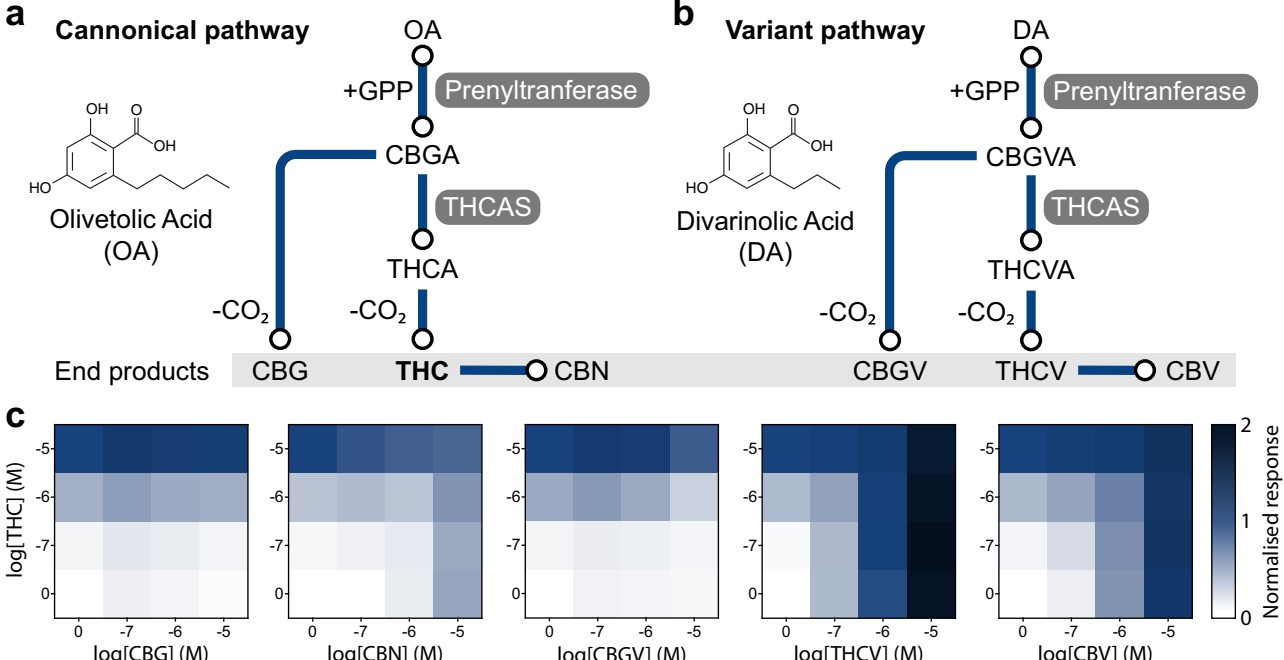

**Fig. 4 | Measuring THC in the presence of pathway by-products. a** Canonical cannabinoid biosynthesis pathway showing the production of cannabigerol (CBG), $\Delta^9$-tetrahydrocannabinol (THC), and cannabinol (CBN) from the olivetolic acid (OA) precursor. **b** Variant cannabinoid biosynthesis pathway showing the production of cannabigerovarin (CBGV), $\Delta^9$-tetrahydrocannabivarin (THCV), and cannabivarin (CBV) from the divarinolic acid (DA) precursor. **c** Response of the CB2 biosensor over a combined range of THC and CBG, CBN, CBGV THCB, or CBV concentrations. Experimental measurements are GFP levels per cell as determined on a plate reader and shown as the mean value from three biological replicates. All data are normalised to no ligand (0) and 10 μM THC (1) CB2 biosensor response.

from microbial fermentation. As THC demonstrated the most favourable signalling qualities with the CB2 biosensor (high sensitivity, large maximum signal output, and wide operational range), we decided to develop and optimise a medium-throughput assay for the screening of this high-value therapeutic compound.

As pH has recently been shown to drastically modulate the signalling of human G protein-coupled receptors in yeast, we first assessed the dose-response of the CB2 biosensor to THC over a range of physiologically relevant conditions[37] (Supplementary Fig. 5a). Increasing the pH of the media from 4 to 7 saw an increase in sensitivity, operational range, and basal activity, with pH 6 providing the most desirable biosensor characteristics (largest and most linear operational range, high sensitivity, and low basal activity). Next, we decided to increase the throughput and accessibility of the CB2 biosensor assay by transferring measurement from flow cytometry to a plate reader and optimising incubation time, cell density, and sample volume (Supplementary Fig. 5b–d). The best results were achieved by resuspending a 24 h saturated culture of the CB2 biosensor in 5 × volume (1:5 dilution) of fresh SC media (pH 6), immediately incubating the cells with THC for 3 h in a 96-deep well plate (30 °C, 700 rpm), and then transferring 300 μL of each well into a black, clear bottom 96-well plate for measurement in a plate reader using bottom-read mode (Fig. 3a).

The final CB2 biosensor plate reader assay demonstrated an ideal biosensor dose-response curve, linear over 3 orders of magnitude between 0.01 and 10 μM of THC, a greater than 6-fold dynamic range, and low error between repeated measurements (StDev < 3 % of maximum signal) (Fig. 3b). As reported THCA titres from yeast are around 10 μM[9], and as the biosensor workflow dilutes the sample by 100-fold, final concentrations of THC from decarboxylated samples of yeast extract should be around 0.1 μM for these strains when used in this assay, making the linear range of the CB2 biosensor highly relevant for this application. Of course, extracted samples can be diluted or concentrated if they fall outside of this range.

## Screening THC in the presence of cannabinoid pathway by-products

Microbial production of cannabinoids has the advantage that bioengineered strains can lack the enzymes that diversify the gateway precursor, CBGA, into the plethora of molecules seen in plants[5]. This simplifies the system over plant extract, as cannabinoids, such as CBD and CBC, can be excluded by simply omitting their synthases, thus reducing the number of cannabinoids that can affect the CB2 biosensor. However, in the decarboxylated extracts of yeast engineered to produce THCA, we still expect to see two other compounds in the canonical cannabinoid biosynthesis pathway which can interfere with THC sensing: CBG, the decarboxylated by-product of the unused gateway cannabinoid, CBGA[9], and cannabinol (CBN), the oxidised product of THC that accumulates over time in high light or oxygen conditions[36] (Fig. 4a).

A further complication of microbial cannabinoid production is an alternative branchpoint in the biosynthetic pathway, creating the variant cannabinoid precursor, cannabigerovarinic acid (CBGVA)[9]. The formation of CBGVA originates from the assembly of GPP with divarinolic acid (DA), an olivetolic acid analogue produced from butanoyl-CoA (an intermediate in the hexanoyl-CoA pathway), due to the promiscuity of at least some pathway enzymes[9]. This can be somewhat overcome with the supplementation of HA and/or OA to push the pathway toward CBGA over CBGVA, however, some variant cannabinoids will always be present from microbial production using complete biosynthesis from sugar feedstocks[9]. From this variant pathway, we expect to see three more cannabinoids in the decarboxylated cell extracts: cannabigerovarin (CBGV), $\Delta^9$-tetrahydrocannabivarin (THCV), and cannabivarin (CBV), which also affect the CB2 biosensor response (Fig. 4b and Supplementary Fig. 6).

To explore the effect of these cannabinoids on the CB2 biosensor response to THC, we set up five two-dimensional dose-response curves, where we tested several THC concentrations in combination with several CBG, CBN, CBGV, THCV, and CBV concentrations over the

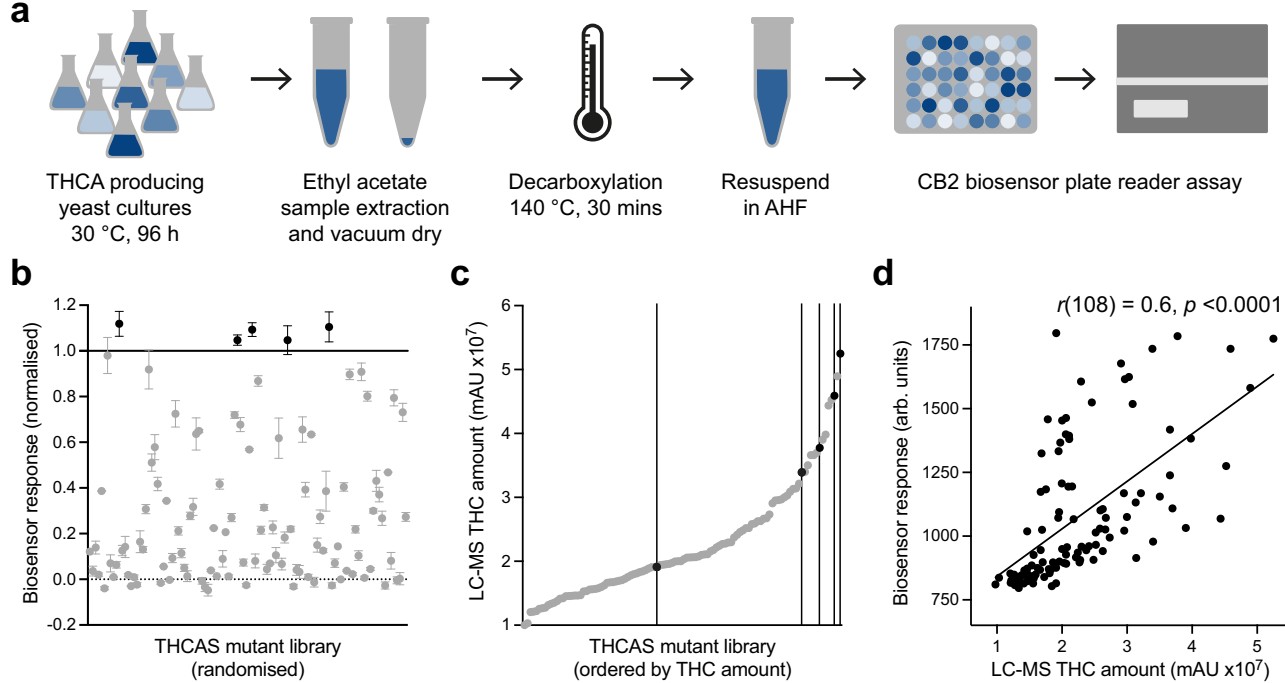

**Fig. 5 | Screening a THCAS mutant yeast library for high producers. a** Workflow for screening THC from microbial production using the yeast CB2 biosensor. AHF, acetonitrile/H$_2$O/formic acid (80/20/0.05 %). **b** Screening a library of 108 THCAS mutant yeast strains for increased THCA production using the CB2 biosensor, highlighting five stains with increased THC titres over the unmutated THCAS (black). Experimental measurements are GFP levels per cell as determined on a plate reader and shown as the mean ± SD from two biosensor measurements of a single extracted sample. Measurements were normalised to no ligand (0, dotted line) and unmutated THCAS control (1, solid line). Samples with StDev > 5 % of the maximum signal were omitted (1/109 samples). **c** Relative quantification of THC from the 108 samples shown in **b** by LC-MS, highlighting the five strains identified as high producers using the CB2 biosensor (black). Experimental measurements are relative THC amounts (mAU) as determined by LC-MS and shown as individual values. **d** Correlation of the CB2 biosensor response with relative quantification of THC by LC-MS, demonstrating a Pearson's correlation coefficient of 0.6, $p$ (two-tailed) < 0.0001. Data are mean values from **b** and individual values from **c** for the 108 decarboxylated cell extracts. A straight-line curve was fitted using GraphPad Prism linear regression fit, $R^2 = 0.36$.

linear range of the CB2 biosensor (Fig. 4c). CBG and CBGV, displayed very little interference of the THC response and so we do not expect these molecules to have much impact when screening microbial samples. CBN and CBV, produced an agonistic response similar to THC, which may result in a slight overestimation of THC concentrations due to their additive effect on the CB2 biosensor output. However, we do not anticipate the accumulation of these products in fresh samples, as decomposition from THC and THCV requires high temperatures (300 °C) or extended periods of time exposed to oxygen[36], which we can control.

THCV resulted in a maximum signal output 2 times greater than THC and starts activating the CB2 biosensor at a concentration 1 order of magnitude lower than THC. As previous reports of cannabinoid production in yeast reported THCA and THCVA were produced in 2:1 ratio[9], the total CB2 biosensor response from decarboxylated cell extract will therefore likely be a contribution from both THC and THCV. This is, nonetheless, informative for pathway engineering as THCV is a direct analogue of THC that uses the same biosynthetic steps. However, the exact ratio of these two cannabinoids will be indistinguishable using the CB2 biosensor, and pathway engineering to shift the ratio of THCV towards THC will require a more quantitative method, such as LC-MS.

### Screening THC from microbial fermentation

To determine whether the CB2 biosensor would be useful for screening microbially produced THC, we next set out to achieve detection from engineered yeast producer cells. First, we established a workflow for extracting cannabinoid samples from yeast and efficiently converting THCA to THC by decarboxylation (Supplementary Fig. 7a). At 140 °C, pure samples of THCA decarboxylate into THC in under

30 min[36]. However, this may vary for complex samples, which will also contain a mixture of other cannabinoids, all contributing to the CB2 biosensor output. Therefore, we designed an experiment to determine the optimal length of time to incubate microbial cell extracts at 140 °C that would produce the greatest CB2 biosensor response.

For this initial test, we created a THCA producing yeast strain (yZ135) that uses an engineered GPP pathway and supplemented olivetolic acid to produce CBGA, which is then transformed into THCA using the *Cannabis sativa* THCA synthase (*Cs*THCAS). The decision to omit the olivetolic acid biosynthesis pathway was made to avoid creating divarinolic acid and subsequent production of the variant cannabinoids. We then grew yZ135 in 30 mL galactose medium (YPG) at 30 °C, 250 rpm for 96 h, feeding with 0.1 mM olivetolic acid and 2 % galactose every 24 h. Cannabinoids were purified by ethyl acetate extraction and samples were vacuum dried and incubated at 140 °C for 0, 20, 25, and 30 min, followed by resuspension in acetonitrile/H$_2$O/formic acid (80/20/0.05 %) and LC-MS for relative quantification (Supplementary Fig. 7b). A trade off was seen between 25 and 30 min where THCA levels were lowest at 30 min, but the highest amount of THC was seen at 25 min, likely due to the decomposition of THC. However, 30 min of heating produced a greater CB2 biosensor response (Supplementary Fig. 7c), and so was chosen as the optimal time for incubation in the final CB2 biosensing workflow (Fig. 5a).

After establishing the sample preparation protocol, we next created a library of yeast strains, based on yZ135, with a range of different THCA titres by introducing an additional copy of *Cs*THCAS where V415 was mutated to any of the 20 canonical amino acids using an NNK codon (yS231 library). We randomly picked 109 of these strains for screening using the CB2 biosensor workflow to see whether we could measure relative differences in production using the CB2 biosensor

and identify any mutants with improved THCA production. This produced high quality data for 108 of the strains (StDev < 5 % from duplicate measurements) that mostly lay within the linear range of the biosensor, identifying 5 strains which displayed a higher signal than a control with an additional copy of an unmutated *Cs*THCAS (yS234) (Fig. 5b).

We then measured the decarboxylated samples by LC-MS to directly quantify relative THC amounts (Fig. 5c). This demonstrated 4/5 hits identified in the CB2 biosensor screen were clustered within the top 15 % of THC producers, while also picking out the top producer. Directly comparing the CB2 biosensor output with relative THC amounts showed a moderate correlation ($r = 0.6$, $p < 0.0001$), demonstrating that the biosensor has promise for enriching libraries for improved THCA producers in large-scale screening experiments (Fig. 5d).

## Discussion

In this study, we developed a yeast GPCR-based biosensor for detecting cannabinoids over wide operational and dynamic ranges, capable of detecting molecules including the therapeutic compounds, $\Delta^9$-tetrahydrocannabinol, cannabidiol, cannabigerol, and cannabichromene. The biosensor was created by expressing the human cannabinoid type 2 receptor in *S. cerevisiae* and coupling this to an easy-to-measure GFP reporter output via a minimised pheromone response pathway.

We extensively characterised the CB2 biosensor response to compounds within the context of microbial cannabinoid production, which suggested a useful potential for screening THC from complex mixtures. We then established a biosensing workflow for detecting THC from microbial production and optimised sample preparation conditions for relative comparison of different producer strains. Finally, we used the CB2 biosensor workflow for screening THCA production from a library of THCAS mutant yeast variants, demonstrating a trend towards higher GFP output with increasing THC titres which could be used to enrich for high producers.

Our work lays the foundation for cheap and scalable screening of microbially produced THC that will accelerate metabolic engineering efforts to compete with traditional plant sources. Sample preparation is amenable to high-throughput approaches and GFP fluorescence can be rapidly measured using a plate reader. While precise quantification may not be achievable using this biosensor, we have demonstrated that high THCA producers can be identified from a large library of yeast strains and the CB2 biosensor has a threshold for THC detection appropriate to this application.

We therefore see the THC biosensing workflow developed here as a method to enrich large libraries of THCA producing variants for downstream analysis by LC-MS, which may not be freely available for large-scale screening efforts or is cost prohibitive to use in this manner. The biosensing protocol reported here relies on lab equipment that typically does not incur additional cost to use (plate reader and vacuum oven), requires cheap reagents and plastic wear (~ $0.1/sample), and only requires one additional step of sample preparation over LC-MS (decarboxylation), which is quick, low cost, and can be scaled.

While the CB2 biosensor proved useful for screening THC from microbial production, the biosensor has limitations. Firstly, we were unable to disentangle the THC biosensor response from the other cannabinoids in the extracts, particularly the variant THCV. To improve on this, the CB2 receptor could be evolved to improve the specificity for THC. G protein-coupled receptors are particularly amenable to directed evolution and established protocols exist for implementing this in yeast[38]. This approach could also be used to specifically target other cannabinoids, whether for point-of-care diagnostics or as a tool for measuring more complex samples, such as plant extract.

In the future, we expect to see more GPCR-based biosensors used as tools for metabolic engineering. The incredible diversity of ligands and stimuli GPCRs can detect provides an unmatched and still largely untapped source of biosensing elements. However, challenges remain. As shown in this work, not all mammalian GPCRs functionally express in yeast, and those that have been previously reported may not work in other settings without the original knowhow. Work is needed to uncover more universal principles for improving the expression, localisation, and coupling of GPCRs in yeast, but progress towards this is underway[37,39].

## Methods

### Strains and cultivation conditions

NEB® Turbo Competent *E. coli* was used for propagating all plasmids and grown at 37 °C in Luria Broth (LB) medium containing the appropriate antibiotics for plasmid selection (ampicillin 100 μg/mL, chloramphenicol 34 μg/mL, or kanamycin 50 μg/mL).

*S. cerevisiae* strain yWS677 (MATα *his3Δ1 leu2Δ0 met15Δ0 ura3Δ0 sst2Δ0 far1Δ0 bar1Δ0 ste2Δ0 ste12Δ0 gpa1Δ0 ste3Δ0 mf(alpha) 1Δ0 mf(alpha)2Δ0 mfa1Δ0 mfa2Δ0 gpr1Δ0 gpa2Δ0*) was used to create all biosensing strains, as described in described in Shaw et al.[16]. After Fig. 1, The CB2 biosensor strain, yWS2345 (yWS677 *ura3::LexO(6x)-pLEU2m-sgGFP-tTDH1-pPGK1-GPA1-tENO1-pRAD27-LexA-PRD-tENO1-URA3; leu2::pCCW12-CB2R-tTDH-LEU2*), and CB2R control strain with a truncated Gpa1, yWS2356 (yWS677 *ura3::LexO(6x)-pLEU2m-sfGFP-tTDH1-pPGK1-tGPA1-tENO1-pRAD27-LexA-PRD-tENO1-URA3; leu2::pCCW12-CB2R-tTDH-LEU2*), were used in all subsequent experiments. The THCA producer strain yZ135 (CEN.PK2-1C *erg9::CTR3p-ERG9; leu2-3,112::His3MX6-GAL1p-ERG19-GAL10p-ERG8; ura3-52::GAL1p*-EfMvaS(A1 10G)-*CYC1t-GAL10p*-EfMvaE-*ADH1t; his3-1::*hphMX4-*GAL1p-ERG12-GAL10p*-IDI1; 308a::*GAL1p-ERG20*(F96W-N127W)-*TDH1t; 1114a::Gal1p*-CsPT4-*TDH1t;* DPOX1::p*LEU2-LEU2-LEU2t; 416d::GAL1p-Cs*THCAS-*ADH1t*) was used for establishing the sample preparation workflow. The THCAS mutant yS231 library (yZ135 YPRCd15c::*GAL1p-Cs*THCAS (V415NNK)-*ADH1t; TRP1p-TRP1-TRP1t*) was used to create a range of THCA producers for screening with the CB2 biosensor. The unmutated THCAS control strain yS234 (yZ135 YPRCd15c::*GAL1p-Cs*THCAS-*ADH1t; TRP1p-TRP1-TRP1t*) was used as a reference in the THCAS mutant screen. *Cs*THCAS was sequenced in all the strains from the yS231 library to identify the codon change at V415, which can be found in Supplementary Table 1.

Yeast were transformed using the lithium acetate protocol by Gietz and Woods[40]. A yeast colony was picked and grown to saturation overnight in YPD. The following morning the cells were diluted 1:100 in 15 mL of fresh YPD in a 50 mL conical tube and grown for 4–6 h to $OD_{600}$ 0.8–1.0. Cells were pelleted and washed once with 10 mL 0.1 M lithium acetate (LiOAc) (Sigma). Cells were then resuspended in 0.1 M LiOAc to a total volume of 100 μL/transformation. 100 μL of cell suspension was then distributed into 1.5 mL reaction tubes and pelleted. Cells were resuspended in 64 μL of DNA/salmon sperm DNA mixture (10 μL of boiled salmon sperm DNA (Invitrogen) + DNA + ddH$_2$O), and then mixed with 294 μL of PEG/LiOAc mixture (260 μL 50 % (w/v) PEG-3350 (Sigma) + 36 μL 1 M LiOAc). The yeast transformation mixture was then heat-shocked at 42 °C for 40 min, pelleted, resuspended in 200 μL of sterile H$_2$O, and plated onto the appropriate synthetic dropout medium.

Yeast extract peptone dextrose (YPD) was used for culturing cells in preparation for transformation: 1 % (w/v) Bacto Yeast Extract (Merck), 2 % (w/v) Bacto Peptone (Merck), 2 % glucose (VWR). All liquid biosensor experiments were performed in synthetic complete (SC) medium with 2 % (w/v) glucose (VWR), 0.67 % (w/v) Yeast Nitrogen Base without amino acids (Sigma), 0.14 % (w/v) Yeast Synthetic Dropout Medium Supplements without histidine, leucine, tryptophan, and uracil (Sigma), 20 mg/L uracil (Sigma), 100 mg/L leucine (Sigma), 20 mg/L histidine (Sigma), and 20 mg/mL tryptophan (Sigma), titrated to the desired pH using NaOH, and filter sterilised. Pre-cultures of the THCA producer yeast strains were grown 2× YP (2 % (w/v) Bacto Yeast

Extract (Merck), 4 % (w/v) Bacto Peptone (Merck)) with 2 % glucose, and production was performed in 2× YP with 2 % galactose (Merck).

## Ligand sensing protocol

All biosensor strains were picked into 500 μL of synthetic complete (SC) media and grown in 2.2 mL 96 deep-well plates at 30 °C in an Infors HT Multitron, shaking at 700 rpm overnight. The next day, saturated strains were then diluted 1:100 into 495 μL of fresh SC fresh media. After 2 h of incubation 5 μL of ligand was added and the cultures were incubated for a further 4 h before measurement. Ligands were dissolved in 100 % ethanol, methanol, acetonitrile, or DMSO (Supplementary Table 2), and the final concentration of solvent used in cultures was 1 % for single ligands and 2 % for double ligands. Cell fluorescence was measured by a SpectraMax plate reader (Molecular Devices) and Attune NxT Flow Cytometer (Thermo Scientific), as indicated in the figure legend. For plate reader measurements, the following settings were used: excitation 485/20, emission 528/20, gain 80, and GFP fluorescence per cell was calculated in Microsoft Excel by dividing fluorescence by $OD_{600}$. For flow cytometry measurements, the following settings were used: FSC 300 V, SSC 350 V, BL1 500 V. Fluorescence data was collected from 10,000 cells for each sample and analysed using FlowJo software, gating for singlets using FSC-A vs FSC-H. No further gating water performed on yeast populations. The gating strategy is described in Supplementary Note 1.

## Growth curves

Single colonies of wildtype BY4741 yeast were grown to saturation overnight in 3 mL YPD. The next day, the yeast cultures were back diluted to an $OD_{600}$ of 0.175, and 99 μL was transferred to a 96-well clear, flat-bottom microplate (Corning). 1 μL of the respective ligand was then added to each well and $OD_{600}$ was the measured over 24 h by a Synergy HT Microplate Reader (BioTek) taking measurements every 15 min with shaking at 30 °C in between readings. Maximum growth rate was then calculated in Microsoft Excel according to the equation $(\ln(OD_{600}(t+3\,h)/OD_{600}(t))/3$, where $t$ is time in hours.

## Yeast cannabinoid culture and sample prep

Cannabinoid producing yeast strains were inoculated into 2 mL 1× YP with 2 % glucose media and allowed to grow at 30 °C, 800 rpm for 24 h. This seed culture was then inoculated into 2 ml 2× YP with 2 % galactose at a 50-fold dilution for 15 h. The secondary seed culture was then back diluted to 0.2 OD $ml^{-1}$ into 2 ml 2× YP with 2 % galactose media and allowed to grow at 30 °C, 800 rpm for 96 h, during which 0.1 mM olivetolic aid and 2 % galactose were added every 24 h. 2 mL culture was then collected, and 1.7 ml culture medium was removed after centrifugation. The cannabinoid products were extracted using a tissue lyser (4 °C, 68 Hz, 90 s turn on/30 s turn off, 16 times) with 600 μL ethyl acetate and 200 μL glass beads (0.5 mm). After extraction, 340 μL upper layer of ethyl acetate was transferred for further steps. 600 μL ethyl acetate were added for a second extraction and the mixture was vortexed at 2500 rpm for 1 min. Another 500 μL of the upper layer was combined with the first extracts and then dried using an Eppendorf Vacufuge Plus. Samples were then heated in a vacuum oven at 140 °C, 30 min (or otherwise specified in the figure legend) and resuspended in acetonitrile/$H_2O$/formic acid (80/20/0.05 %) for biosensor experiments and subjected to LC-MS quantification.

## LC-MS quantification of microbial cannabinoid samples

The extracted cannabinoid samples were analysed using LC-MS (1260 infinity liquid chromatograph coupled to Agilent 6470 tri quadrupole mass spectrometer) equipped with reverse phase C18 column (InfinityLab Poroshell 120 EC-C18, 3.0 × 100 mm 2.7-Micron, Agilent). The mobile phase was set as solvent A (Water with 0.05 % formic acid) and solvent B (acetonitrile with 0.05 % formic acid). The compounds were separated via gradient elution as follows: linearly

increased from 30 % B to 40 % B in 3.0 min, then increased to 80 % B in 4.8 min, increased from 80 % B to 97 % B in 3.0 min, held at 97 % B for 2.0 min, decreased from 97 % B to 30 % B in 0.1 min, and held at 30 % B for 2.1 min. The flow rate was held at 0.5 ml $min^{-1}$ and total liquid chromatography run time was 15.0 min. The sample panel and column compartment were set at 15 °C and 40 °C, respectively. The tri quadruple mass spectrometer was set as follows: gas temperature 300 °C, gas flow 5 L $min^{-1}$, nebuliser 45 psi, sheath gas temperature 250 °C, sheath gas flow 11 L $min^{-1}$. Electrospray ionisation was conducted in the negative/positive ion mode and capillary voltage of 3500 V as used. The data files were processed with Agilent MassHunter Qualitative Analysis software.

## Optimised CB2 biosensor assay for screening microbially produced THC

The CB2 biosensor strain, yWS2345, was inoculated into synthetic complete (SC) media (pH 6.0) (1 assay = 10 mL pre-culture volume) and grown at 30 °C, 250 rpm for 24 h. The next day, cells were pelleted in large bench top centrifuge (2000 × $g$, 3 min), and resuspend in 5 × volume of SC medium (pH 6.0) (1 assay = 50 mL final volume). 495 μL of cells were transferred into each well of a 96-deep well plate (2.2 mL conical bottom) and 5 μL of the decarboxylated cell extract were added to each well and incubated for 3 h at 30 °C, 800 rpm. 300 μL of the cells were then transferred into a Costar 96 black plate and measured without cover. The $OD_{600}$ and GFP fluorescence intensity measurements (488 nm excitation and 510 nm emission) were performed using a Tecan Infinite 200 PRO plate reader.

## Statistics and reproducibility

Unless otherwise stated, all data was analysed in Prism (GraphPad). Error bars represent the standard deviation of the mean. The respective number of replicates are given in the figure legend and all replicates are included in the manuscript. All presented curve fittings were generated in Prism (GraphPad).

## Reporting summary

Further information on research design is available in the Nature Research Reporting Summary linked to this article.

## Data availability

Nucleotide sequence data of all G protein-coupled receptors used in this study are included in Supplementary Table 3. Nucleotide sequence data for all other GPCR biosensor components were previously reported in Shaw et al. (2019)[16]. Nucleotide sequence data for all cannabinoid producing strains were previously reported in Luo et al. (2019)[9]. The CB2 biosensor strain will be made available from the corresponding author. Strains producing controlled substances or direct precursors of controlled substances can only be provided to laboratories/institutions with appropriate approvals and licences. Individual data points for all graphs are provided as source data with this paper. Source data are provided with this paper.

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

## Acknowledgements

X.Luo and Y.Z. would like to thank the support of the National Key Research and Development Programme of China (2018YFA0903200). W.M.S. and T.E. would like to thank the support of the URKI Biotechnology and Biological Sciences Research Council (BBSRC) awards BB/M503381/1 and BB/R002614/1. A.S.K. acknowledges funding from National Science Foundation (NSF) grant CCF-2027045, Department of Defence (DoD) Vannevar Bush Faculty Fellowship N00014-20-1-2825, and the W.M. Keck Foundation.

## Author contributions

W.M.S., Y.Z., X.Luo, and T.E. designed the experiments. W.M.S., Y.Z., and X.Lu performed the experiments. W.M.S., and Y.Z. performed the data analysis. W.M.S., Y.Z., G.L., A.S.K., X.Luo, and T.E. interpreted the results. W.M.S. wrote the manuscript. All authors reviewed and approved the final manuscript.

## Competing interests

X.Luo has a financial interest in Demetrix. A.S.K. is a scientific advisor for and holds equity in Senti Biosciences and Chroma Medicine, and is a co-founder of Fynch Biosciences and K2 Biotechnologies. T.E. is paid as a scientific consultant for Replay and is a scientific advisor for and holds equity in Modern Synthesis. X.Luo, A.S.K., and T.E. declare no further financial and non-financial competing interests. The remaining authors declare no financial or non-financial competing interests.
