## [Peer Review File · Nature Communications]

REVIEWERS' COMMENTS

Reviewer #1 (Remarks to the Author):

As I am a reviewer of the former version of this manuscript, I will firstly comment on the rebuttal put forward by the authors to my initial questions and suggestions for improvements.

Based on the extensively redesigned study, esp. the direct comparisons of biosensor read-outs and quantifications using analytical chemistry of decarboxylated samples now makes the study more clear and compelling. Additionally, the THCAS library is now extended providing higher resolution of the CB2R biosensing capabilities. With these new additions, the authors have addressed my initial feedback in a much improved revised manuscript.

Next, based on the revised manuscript it is now even more evident that the testbed presented in this study is really complex, incl. alternative pathway branch points, promiscuous enzymes creating variant precursors (DA vs OA, CBGA vs CBGVA) as well as agonistic effects of some pathway precursors on CB2R. This is also reflected in the relative low correlation coefficient between MS and biosensor data ($r = 0.6$). Yet as a prescreen, the CB2R assay will increase the hit rate when searching for optimised pathway variants, and I honestly hope the authors can put the biosensor to use for the engineering purposes conducted at Demetrix and other cannabinoid companies taken this important class of molecules to market.

Before accepting the revised manuscript, I would really like the authors' feedback on one last comment arising from the new data presented. As no patent application seems to have been filed (as viewed from the conflict statement), I am puzzled why the authors do not present any sequencing data on optimised THCAS mutants. Ideally, the authors should be able to tell which of the 19+1 aa THCAS mutants showed the highest THC titers, and furthermore show replicate biosensor read-outs of a few selected sequence-validated THCAS mutants. With 108 colonies screened in a 20 aa possibility-window at V415, a few mutants with identical non-synonymous mutations must have been sampled, and this would be important information for the specialised readership interested in following up on the CB2R and THCAS work from this study. Besides a response to this comment, I have no further comments to the revised study.

Reviewer #3 (Remarks to the Author):

All of my concerns are properly addressed in the revised version of the manuscript.

Response to reviewer comments for Shaw et al.

We have now modified the manuscript to include additional experimental work, as suggested by Reviewer #1, to determine the codon at V415 for the entire yS231 THCAS mutant library. This is now included as Supplementary Table 1.

Our response to the reviewer comments are below.

Reviewer #1 (Remarks to the Author):

As I am a reviewer of the former version of this manuscript, I will firstly comment on the rebuttal put forward by the authors to my initial questions and suggestions for improvements.

Based on the extensively redesigned study, esp. the direct comparisons of biosensor read-outs and quantifications using analytical chemistry of decarboxylated samples now makes the study more clear and compelling. Additionally, the THCAS library is now extended providing higher resolution of the CB2R biosensing capabilities. With these new additions, the authors have addressed my initial feedback in a much improved revised manuscript.

We thank the reviewer for their time and continued efforts to improving our manuscript.

Next, based on the revised manuscript it is now even more evident that the testbed presented in this study is really complex, incl. alternative pathway branch points, promiscuous enzymes creating variant precursors (DA vs OA, CBGA vs CBGVA) as well as agonistic effects of some pathway precursors on CB2R. This is also reflected in the relative low correlation coefficient between MS and biosensor data ($r = 0.6$). Yet as a prescreen, the CB2R assay will increase the hit rate when searching for optimised pathway variants, and I honestly hope the authors can put the biosensor to use for the engineering purposes conducted at Demetrix and other cannabinoid companies taken this important class of molecules to market.

Before accepting the revised manuscript, I would really like the authors' feedback on one last comment arising from the new data presented. As no patent application seems to have been filed (as viewed from the conflict statement), I am puzzled why the authors do not present any sequencing data on optimised THCAS mutants. Ideally, the authors should be able to tell which of the 19+1 aa THCAS mutants showed the highest THC titers, and furthermore show replicate biosensor read-outs of a few selected sequence-validated THCAS mutants. With 108 colonies screened in a 20 aa possibility-window at V415, a few mutants with identical non-synonymous mutations must have been sampled, and this would be important information for the specialised readership interested in following up on the CB2R and THCAS work from this study. Besides a response to this comment, I have no further comments to the revised study.

The strains we created in the THCAS mutant library by randomising the codon at V415 were made to create diversity in THCA production. This library was then used assess the performance of the biosensor compared to LC-MS. Ideally this would have also produced

greatly improved production strains. However, the best strains were only marginally better than the wild type THCAS. As we did not see this as a great improvement in titres, and likely not a very useful codon to mutate for future studies, we decided not to sequence the library. However, we agree the absence of this data left an incomplete story. To address this point, we have now sequenced all 108 strains in the THCAS mutant library to determine the codon at amino acid 415. This has now been included as Supplementary Table 1 alongside a summary of the relative amount of THC and the duplicate biosensor response for quick reference. We hope this now completes the story and can be used by others following on from this work.

We also thank the reviewer for pointing out that our author conflicts statement was not complete. We've updated this now accordingly.

Reviewer #3 (Remarks to the Author):

All of my concerns are properly addressed in the revised version of the manuscript.

We thank the reviewer for their time and commitment to improving our manuscript.